# Targeted Isolation of Prenylated Flavonoids from *Paulownia tomentosa* Fruit Extracts via AI-Guided Workflow Integrating LC-UV-HRMS/MS

**DOI:** 10.3390/metabo15090616

**Published:** 2025-09-17

**Authors:** Tomas Rypar, Lenka Molcanova, Barbora Valkova, Ema Hromadkova, Christoph Bueschl, Bernhard Seidl, Karel Smejkal, Rainer Schuhmacher

**Affiliations:** 1Department of Chemistry and Biochemistry, Mendel University in Brno, Zemedelska 1, CZ 613 00 Brno, Czech Republic; tomas.rypar@mendelu.cz; 2Institute of Bioanalytics and Agro-Metabolomics, Department of Agricultural Sciences, BOKU University, Konrad-Lorenz-Str. 20, 3430 Tulln, Austria; christoph.bueschl@boku.ac.at (C.B.); bernhard.seidl@boku.ac.at (B.S.); 3Department of Natural Drugs, Masaryk University, Palackeho 1946/1, CZ 612 00 Brno, Czech Republic; molcanoval@pharm.muni.cz (L.M.); smejkalk@pharm.muni.cz (K.S.); 4Core Facility Bioactive Molecules: Screening and Analysis, BOKU University, Konrad-Lorenz-Str. 20, 3430 Tulln, Austria

**Keywords:** bioactive compounds, geranylated flavonoids, prenylated polyphenols, specialized metabolites, untargeted metabolomics

## Abstract

Objectives: This study presents a versatile, AI-guided workflow for the targeted isolation and characterization of prenylated flavonoids from *Paulownia tomentosa* (Thunb.) Steud. (Paulowniaceae). Methods: The approach integrates established extraction and chromatography-based fractionation protocols with LC-UV-HRMS/MS analysis and supervised machine-learning (ML) custom-trained classification models, which predict prenylated flavonoids from LC-HRMS/MS spectra based on the recently developed Python package AnnoMe (v1.0). Results: The workflow effectively reduced the chemical complexity of plant extracts and enabled efficient prioritization of fractions and compounds for targeted isolation. From the pre-fractionated plant extracts, 2687 features were detected, 42 were identified using reference standards, and 214 were annotated via spectra library matching (public and in-house). Furthermore, ML-trained classifiers predicted 1805 MS/MS spectra as derived from prenylated flavonoids. LC-UV-HRMS/MS data of the most abundant presumed prenyl-flavonoid candidates were manually inspected for coelution and annotated to provide dereplication. Based on this, one putative prenylated (C5) dihydroflavonol (1) and four geranylated (C10) flavanones (2–5) were selected and successfully isolated. Structural elucidation employed UV spectroscopy, HRMS, and 1D as well as 2D NMR spectroscopy. Compounds **1** and **5** were isolated from a natural source for the first time and were named 6-prenyl-4′-O-methyltaxifolin and 3′,4′-O-dimethylpaulodiplacone A, respectively. Conclusions: This study highlights the combination of machine learning with analytical techniques to streamline natural product discovery via MS/MS and AI-guided pre-selection, efficient prioritization, and characterization of prenylated flavonoids, paving the way for a broader application in metabolomics and further exploration of prenylated constituents across diverse plant species.

## 1. Introduction

Flavonoids are specialized plant metabolites that play key roles in environmental interactions. For instance, they provide protection against UV irradiation, pathogens, or herbivores, and attract pollinators [1]. These compounds are also important dietary components of human food with various biological activities: they can, for example, be antioxidative, anti-inflammatory [2], antimicrobial [3], anticancerogenic [4], or show protective effects against chronic diseases [5]. Prenylated flavonoids are also recognized as key active constituents in many traditional Chinese medicines and functional foods.

Prenylation of flavonoids, e.g., through attachment of a prenyl (C5) or geranyl (C10) side chain, has been reported to enhance several bioactivities, including antioxidant [6], α-glucosidase inhibitory [7], bone [8], and muscle maintaining [9], and anti-inflammatory effects [10]. However, unlike their unsubstituted counterparts, prenylated flavonoids are found predominantly in certain plant families, such as Fabaceae and Moraceae [11], which possess diverse prenyltransferases. These enzymes facilitate the attachment of prenyl groups to various core structures such as flavonoids, xanthones, and coumarins, and exhibit distinct regio- and substrate specificities [12]. Many other prenylated compounds can be found across different species [13].

Natural products have long inspired drug discovery [14,15]. However, the high chemical complexity of plant extracts often leads to repeated isolation of already known compounds, resulting in a loss of time and resources. Therefore, dereplication, i.e., the discrimination between known and novel constituents, is an essential task in discovery-driven studies [16,17]. Numerous databases [18] and advanced instrumentation, particularly LC-HRMS/MS, support this process by providing structural insights into complex mixtures and efficient natural product discovery [19].

A major bottleneck in LC-HRMS/MS analysis is still the annotation and interpretation of fragmentation spectra. A key limitation originates from the scarcity of available experimental reference spectra. Nowadays, machine learning enables overcoming these challenges by using, for example, in silico MS/MS spectra predictions [20,21,22,23,24], structure inference from fragmentation data [25,26,27,28,29,30], or improved spectral matching via automated pattern recognition [31,32,33,34]. Other AI-guided tools for rapid classification and prediction of bioactive scaffolds [35], steroid-like compounds [36], or bioactivity prediction [37] have also been reported. These tools are essential for dereplication in high-throughput analysis of MS/MS spectra and annotation [38]. Although their performance steadily increases, these methods are still often constrained by the limited coverage and quality of (public) spectral libraries [39]. Therefore, rule-based or semi-automated annotation strategies remain valuable, as they can provide precise and reliable annotation and dereplication, particularly for some compound classes, which are only scarcely present in databases [40,41].

While various approaches using full scan LC-HRMS (MS1) data, such as the use of mass defect filters [42], Van Krevelen diagrams (i.e., H/C and O/C ratios) [43], or retention time filters [44], have been demonstrated to select features belonging to a specific compound class of interest, they remain less specific than strategies that utilize fragmentation data containing structural information. Several studies have focused on the discovery of prenylated flavonoids using MS/MS and MS^n^ data acquisition strategies, particularly in plant genera such as *Artocarpus* [45] and *Glycyrrhiza* [42,46]. These approaches typically rely on proposed or known fragmentation rules or neutral loss screening methods [47]. Although they have proven to be very helpful in the search for metabolites of interest, they are labor-intensive and time-consuming, especially in untargeted studies. Moreover, they often depend on only a few diagnostic fragments while ignoring much of the spectral information. In addition, the acquisition of sufficient MS^n^ data per chromatographic peak can be constrained by complex sample composition, limited scan rates of the mass spectrometer, and narrow chromatographic peaks, especially with higher mass resolving power settings. In order to close this gap, machine learning (ML)-based tools can be trained for spectra classification models using, e.g., our custom-made AnnoMe Python package (v1.0) and MS2DeepScore representation of MS/MS spectra for the prediction of prenylated flavonoids [48].

Here, we present a versatile high-throughput workflow that utilizes an optimized extraction protocol for prenylated constituents in combination with LC-UV-HRMS/MS analysis with common data-dependent acquisition and a custom-trained machine-learning classification model to predict prenylated flavonoids from MS/MS spectra. Our workflow enables acquisition of a high number of MS/MS spectra and their classification in a high-throughput manner and the efficient prioritization of fractions or prenylated constituents and streamlines their characterization and targeted isolation from complex plant matrices.

The primary aim of this study is to integrate our previously published work on extraction protocols, fractionations, and UV-spectroscopy [49,50] with LC-HRMS/MS analysis and ML-based structure class prediction into a customized workflow enabling efficient prediction, prioritization, and isolation of prenylated flavonoids from complex plant extracts. We aimed to: (i) demonstrate the utility of ML-based classification in reducing the chemical complexity of LC-HRMS/MS datasets; (ii) streamline the isolation process by dereplication to avoid re-isolation of known compounds and enable structural characterization of novel prenylated flavonoids; and (iii) contribute to the expansion of public spectral repositories. By achieving these objectives, this study provides a robust workflow that can be readily adapted and integrated into common untargeted metabolomics pipelines, supporting not only natural product discovery but also broader metabolomics investigations.

## 2. Materials and Methods

### 2.1. Chemicals

ELGA water was obtained from an ELGA Purelab Ultra-AN-MK2 system (Veolia Water; Vienna, Austria). LC-MS grade acetonitrile (ACN), methanol (MeOH), and formic acid (FA) were obtained from Sigma-Aldrich (Steinheim, Germany, and Vienna, Austria). Gradient grade ACN and MeOH for HPLC separations were purchased from Sigma-Aldrich or VWR International (Fontenay-sous-Bois, France). Reference standards used for MS/MS library generation are listed in Appendix A.

### 2.2. Sample Preparation

The green immature fruits of *Paulownia tomentosa* (Thunb.) Steud. (Paulowniaceae), characterized by a sticky resin on their surface, were collected at the Luzanky city park in Brno, Czech Republic, in late October 2024, and processed on the day of harvest. This period was found to be optimal for harvesting and extracting geranylated flavonoids in central Europe [51]. Overall, 9.13 kg of fruits were harvested and successively macerated three times for 24 h in ethanol (3 × 12.5 L; the last extraction was assisted by sonication for 1 h) to yield 190.9 g of extract. Further liquid-liquid extraction was applied as schemed in Appendix A. Briefly, a part of the dried ethanolic extract (175.9 g) was dissolved in approximately 2.2 L of 90% methanol (*v*/*v*) and extracted four times with approx. 1.2 L of hexane to obtain the hexane portion (12.5 g). Methanol was evaporated, and the rest of the water part was diluted with approx. 2 L of water and extracted four times with approx. 1.2 L of chloroform to obtain the chloroform portion (38.4 g). The water part was subsequently extracted four times with approx. 1.2 L of ethyl acetate to give the ethyl acetate portion (15.6 g) and the water portion. The chloroform portion was further separated by column chromatography on silica gel with a particle size of 40–63 μm (Merck, Rahway, NJ, USA) and methanol:toluene:chloroform (3/5/92, *v*/*v*/*v*) mobile phase composition. After collecting about 60 fractions (125 mL each), the polarity of the mobile phase was increased to support further separation. Another approx. 30 fractions were collected, and finally, the column was washed with pure methanol. A total of 109 fractions were obtained. These were separated and analyzed with TLC, and fractions with similar profiles were merged into 16 fractions. 13 of these were selected for further analysis: fraction 1 (38 mg), fraction 2 (152 mg), fraction 3 (311 mg), fraction 4 (1033 mg), fraction 5 (5358 mg), fraction 6 (3540 mg), fraction 7 (2099 mg), fraction 8 (7774 mg), fraction 9 (1403 mg), fraction 10 (490 mg), fraction 11 (1052 mg), fraction 12 (3691 mg), and fraction 13 (8099 mg). All fractions were dried under nitrogen flow, subsequently under vacuum, and lyophilized to remove solvent residues and stored in a dark place at 8 °C.

### 2.3. General Experimental Procedures

Analytical HPLC measurements were carried out with an Agilent 1100 chromatographic system equipped with an Agilent 1100 Series diode array detector (Agilent Technologies, Santa Clara, CA, USA) to provide a link between laboratories and to obtain UV spectra of isolated compounds. The UV spectra were obtained from the apex of the corresponding chromatographic peak. The same gradient elution for HPLC and LC-HRMS analyses was used (Section 2.4). 1D and 2D NMR spectra were acquired on a JEOL ECZR 400 MHz NMR spectrometer (JEOL, Tokyo, Japan). The residual solvent signals of CDCl_3_ or DMSO-*d*_6_ were used for reference. CDCl_3_ and DMSO-*d*_6_ were purchased from Merck KGaA, Germany. Semi-preparative RP-HPLC was performed using a Dionex UltiMate 3000 HPLC System with a fraction collector (Thermo Fischer Scientific, Waltham, MA, USA) and an Agilent 1100 chromatographic system (Agilent Technologies, Santa Clara, CA, USA) by manual collection of fractions. The semi-preparative HPLC separations were performed with an Ascentis RP-Amide, 25 cm × 10 mm, particle size 5 μm or an Ascentis C18, 25 cm × 10 mm, particle size 5 µm (Supelco, Bellefonte, PA, USA). For analytical separation on TLC plates, silica gel 60 F_254_ (20 × 20 cm, 200 μm, Merck, Rahway, NJ, USA) was used.

### 2.4. LC-HRMS/MS Analysis

Fractions prepared by chromatographic fractionation of the chloroform portion of the ethanolic extract were analyzed. First, 1 mg of dried extract from each fraction was weighed into a glass vial and dissolved in 1 mL of MeOH and further diluted with 3 mL to form a 0.25 mg/mL solution. Samples were injected in a volume of 2 μL and analyzed by high-performance liquid chromatography with UV-Vis absorption detection coupled with the high-resolution mass spectrometry (LC-UV-HRMS) method. A UHPLC Vanquish Horizon system (Thermo Scientific, Bremen, Germany) with a variable wavelength detector was used for chromatographic separation with a versatile 47 min long reversed-phase chromatography method using a 100 mm × 2.1 mm, 2.7 µm analytical HPLC column Ascentis Express RP-Amide (Sigma-Aldrich, St. Louis, MO, USA) and linear gradient elution with ACN (B) and water (A), both acidified with FA (0.1%, *v*/*v*). The gradient started at 10% of B at 0 min and ended at 100% of B at 36 min, followed by a hold up to 40 min and re-equilibrated between 40.1 and 47 min at 10% of B. The flow was set to 0.3 mL/min. The column was kept at 40 °C and the autosampler at 10 °C during the analysis. The Vanquish variable wavelength detector was used to collect data at wavelengths of 275, 293, and 348 nm. An Orbitrap IQ-X (Thermo Scientific, Bremen, Germany) equipped with a heated electrospray ionization source and coupled to the chromatographic system was utilized for data acquisition using a deep scan AcquireX workflow with four DDA iterations. Briefly, mass spectra were recorded in full scan profile mode with a resolving power setting of 120,000 FWHM at *m/z* 200 in negative ionization mode for the generation of an exclusion list from blank samples and inclusion lists with detected features from the analytical fractions. Four successive injections of the same analytical fraction were performed to maximize the number of fragmented precursor ions from the inclusion list with iteratively adapted exclusion and inclusion lists. The fragment spectra were recorded in data-dependent mode, first acquiring full scan profile spectra with a resolving power setting of 60,000 FWHM at *m/z* 200 and a scan range of *m/z* 100–1000 with a normalized AGC target of 50% (2 × 10^5^) and a maximum injection time of 118 ms. Then, the two most intense ions were isolated in the quadrupole analyzer (isolation window of *m/z* 1.2), and fragment spectra were recorded for both ions with higher-energy collisional dissociation (HCD) at 20, 30, and 40 eV (absolute) in negative ionization mode and a resolving power setting of 15,000 FWHM, AGC target 25% (1.25 × 10^4^), maximum injection time 132 ms, and dynamic exclusion of 6 s after a maximum of 6 precursors were triggered. The HESI source was utilized in negative ionization mode with a capillary voltage of 2.5 kV. Sheath, auxiliary, and sweep gases were set to 50, 5, and 0 arbitrary units, respectively. Capillary temperature was set to 275 °C, and the probe heater was kept at 300 °C. The RF lens level was set to 55%.

### 2.5. Isolation of Target Prenylated Compounds

Based on LC-UV-HRMS/MS analysis, fractions containing potentially new prenylated compounds were selected and subjected to further separation by semi-preparative HPLC using gradient or isocratic elution with different ratios of ACN or MeOH and 0.2% FA or water (5 mL/min) and either an Ascentis RP-Amide or Ascentis C18 semi-preparative HPLC column. Subfractions were collected using a fraction collector or manually based on the UV-detector response (*λ* 254, 280, and 350 nm).

Fraction 2 (152 mg) was separated by semi-preparative RP-HPLC using gradient elution (60–85% ACN, 20 min, RP-Amide column) to obtain compounds **5** (0.8 mg, RT 23.79 min) and **3** (RT 26.50 min). Part of this sample was also separated and collected manually, yielding compound **3** (64.9 mg in total).

Part of the fraction 4 (approximately 300 mg) was subjected to semi-preparative RP-HPLC separation using gradient elution (55–85% ACN, 20 min, RP-Amide column) to isolate compounds **4** (13.2 mg, RT 22.73 min) and **2** (41.5 mg, RT 25.99 min). Several attempts to purify compound **2** from another coeluting compound at RT 26.18 min, including variations in solvents, column type, temperature, and preparative TLC, were unsuccessful under the tested conditions.

Compound **1** (4.9 mg, RT 18.31 min) was obtained by repeated purification of fraction 7 (processed amount was approx. 1100 mg) using gradient elution (50–95% ACN, 20 min, RP-Amide column) with automatic collection of the fractions (this separation was performed without addition of FA) and subsequent isocratic elution (50% ACN, C18 column) with manual collection of fraction.

### 2.6. Generation of In-House Spectral Library

From previously isolated geranylated flavonoids and reference standards of flavonoids (Appendix A, stock solutions were prepared. For each substance, 1 mg was dissolved in 1 mL of MeOH or a 1/1 mix of MeOH/DMSO (*v*/*v*) when poor solubility in MeOH was observed. The stock solutions were further diluted by MeOH to form 1 μg/mL of the measurement solutions. For the generation of spectral data for our in-house library, these solutions were analyzed by LC-HRMS/MS under standardized conditions using a 20-min long linear gradient method (MeOH and water both acidified with 0.1% FA *v*/*v*) with a C18 reversed-phase chromatographic column (3.5 μm, 2.1 mm × 150 mm, XBridge^®^ Waters, Milford, MA, USA). Then, 2 μL aliquots of the respective measurement solutions were injected, and full scan mass spectra in fast polarity switching mode were collected to evaluate suitable adducts for targeted HRMS/MS data acquisition for the following injections. For each compound, at least two ion species (most commonly protonated and deprotonated, in positive and negative ionization mode, respectively) were selected in full scan measurements. Moreover, intensities of the selected adducts were evaluated and used to adjust the injection volume in the subsequent MS/MS measurements. The aim was to achieve an intensity of approximately 10^7^ counts for the precursor ion in consecutive scans. Separate injections for each adduct were used to acquire fragmentation spectra with 11 different collision energies (HCD: 10, 20, 30, 40, 50, 60, 70, 80, 90, and 100, and stepped 20, 45, and 70 eV; CID: 10, 20, 30, 35, 40, 50, 60, 70, 80, 90, and 100 eV in absolute values) using targeted HRMS/MS acquisition (tMS2 mode) with a specified RT window and precursor mass. At least 5 fragment spectra of each adduct and collision energy were required for further processing. Collected data were processed by an adopted automatic workflow using the RMassBank R package (v3.18.0) [52] and resulted in an MS/MS library in MassBank format. The library was converted to the mgf format and further used for compound annotation using the MZmine software (v4.5.20) (see below).

### 2.7. Data Analysis

#### 2.7.1. Raw Data Processing

The LC-HRMS/MS raw files were converted to the mzML format with msConvert from the ProteoWizard toolbox (v3.0.21292) [53], and the MS/MS precursor *m/z* was corrected. QC samples (mix of reference standards) injected periodically during the sequence were evaluated for retention time stability, mass accuracy, EIC peak intensity, and area precision over the time of the measurement sequence using an in-house R script. The converted files were further processed using MZmine (v4.5.20) [54] with parameters provided in the batch file (Appendix A). Briefly, features with at least three scans above a defined threshold were detected within an *m/z* window of 10 ppm and further joined across all samples within a retention time window of 0.07 min and an *m/z* window of 8 ppm. Only features with peak width between 0.03 and 0.2 (assessed at FWHM of the peak) and MS/MS spectra were retained. The final feature table containing peak areas and metadata (ID, *m/z*, RT, annotations, etc.) was exported as a csv file. The results were also exported in the mgf format for compound annotation with SIRIUS software (v6.1) [25]. The Python package AnnoMe [48], which provides a framework for building machine learning-based classifiers of specific compound classes, was utilized. For this, reference MS/MS spectra of 64 prenylated flavonoids were labeled as “relevant” spectra, while MS/MS spectra of non-prenylated flavonoids and many other compound classes, as well as experimental MS/MS spectra of a wheat-ear extract, were labeled as “other” spectra. The wheat methanolic extract was shown to be important in lowering the false positive rate [48]. Then, the AnnoMe package was tasked with distinguishing between spectra of the two classes, “relevant” and “other”. The performance of the classification models was monitored using 10-fold cross-validation. Readers can find further details in the technical paper [48]. The obtained classification models were subsequently used to predict the “relevant” and “other” labels for the unknown compounds of the plant extracts that are described here to categorize these as likely prenylated flavonoids (i.e., relevant) or other compounds. Another mgf file was exported to be used for feature-based molecular networking in the MetGem software [55] (v1.5.2.) with the following settings: minimum cosine score 0.7, minimum of 4 matched peaks with *m/z* tolerance of 0.02 Da, keeping peaks with *m/z* values higher than 50, maximum neighbors 10, maximum connected components 1000, scale 30, gravity 1. The final network was exported and further processed in Cytoscape [56] (v3.10.3) for final visualization.

#### 2.7.2. Compound Identification and Annotation

For annotation purposes, the confidence system presented by Blaženović and colleagues [57] was adopted. For this, the reference standards were measured under the same analytical conditions as the chloroform fractions (Section 2.4). Subsequently, MS/MS spectra, *m/z,* and retention times of the reference compounds were matched with the features obtained in the plant extract samples, resulting in their identification (annotation confidence level 1). The fragment spectra were matched against an in-house and a whole GNPS spectral library, resulting in an annotation confidence level of 2. Furthermore, SIRIUS software (v6.1.0) was used for classification [58] and annotation (confidence level 3).

#### 2.7.3. Manual Evaluation of Classification Results

Fragment spectra from LC-UV-HRMS/MS analysis of the plant extracts were manually checked using MZmine and Freestyle software (v1.8) according to the following rules. Features with maximum intensity of the precursor ion lower than 10^5^, often providing low-quality MS/MS spectra, were removed. Features not aligned across at least 5 samples were also filtered out. Precursor ions annotated through spectral library matching with the cosine similarity score greater than 0.8 and containing at least five fragment ions were considered confirmed prenylated flavonoids (PF confirmed), or non-prenylated in other cases (not PF). Additionally, precursor ions with cosine similarity scores between 0.7 and 0.8 and at least five fragments were classified as presumably prenylated flavonoids and further evaluated manually when feasible. Fragmentation rules and mechanisms present in the literature [40,46,59,60,61] and those observed from our reference standard spectra were further utilized. For several hundreds of detected features, the MS/MS spectra were inspected and classified. Features were assigned as prenylated flavonoids (PF confirmed) when the spectral evidence was strong. In cases of lower confidence, due to poor MS/MS quality or missing diagnostic fragments, features were labeled as presumed PF. Many other features were assigned as not prenylated flavonoids (not PF). Instances with inconclusive evidence or insufficient knowledge were labeled as uncertain. Only features manually classified as “PF confirmed” and “not PF” were used to evaluate the performance of the AnnoMe-derived compound classification.

Additionally, the CANOPUS classification, specifically ClassyFire’s “most specific class” field, was searched for the keywords “prenyl” and “glyco”, as these terms commonly appeared in the classification of geranylated/prenylated flavonoid standards and glycosides, respectively. If the classification output contained “prenyl,” the compound was considered prenylated. Otherwise, it was considered non-prenylated. A similar approach was used in the case of the classification with glycosylated constituents.

## 3. Results

### 3.1. Workflow Overview

The ethanolic extract of the immature fruit of *P. tomentosa* was separated by liquid-liquid extraction (Appendix A) into four portions to reduce sample complexity and to obtain the chloroform portion, enriched in hydrophobic prenylated constituents, based on our established procedure [49,50]. The chloroform portion was subjected to additional separation using column chromatography on silica gel to further reduce sample complexity and concentrate the presumed target compounds. To support our natural product discovery efforts, we incorporated LC-UV-HRMS/MS analysis together with a machine learning-based classifier into our workflow to facilitate prioritization of fractions and compounds, further characterization, dereplication, and streamlining of isolation of prenylated flavonoids, as outlined in Figure 1. A widely used data-dependent acquisition (DDA) method, AcquireX deep scan workflow, was employed to collect comprehensive fragmentation spectra without relying on specialized acquisition strategies that could limit downstream data exploration. In addition to conventional data processing with MZmine and SIRIUS for feature detection, alignment, library matching, annotation, classification, or molecular networking, we trained a machine-learning model with MS/MS spectra of known prenylated flavonoids to predict if the compounds in the extracts are likely prenylated flavonoids or not. This enabled rapid filtering of the vast chemical space typically encountered in LC-HRMS data obtained from plant material. Candidate compounds were further prioritized before isolation based on their abundance and coelution patterns with other features, considering both UV and HRMS data. When isolation was feasible, the prenylated targets were examined in the molecular network, and their fragment spectra were manually evaluated to propose a potential structure and perform dereplication via SciFinder before purification with semi-preparative HPLC. Structural elucidation of isolated compounds utilizing UV spectroscopy, LC-HRMS, and NMR confirmed the effectiveness of the workflow.

### 3.2. Prenylated Constituents in Paulownia tomentosa

The LC-UV-HRMS/MS analysis of all 13 chloroform fractions resulted in a total of 2687 detected features. The complete feature list, including metadata (e.g., annotation, classification) and peak areas, is provided in Appendix A). All listed features fulfilled three criteria: (i) have MS/MS spectra available, (ii) detected in at least 5 of the 65 injections, and (iii) have an ion intensity of at least 1 × 10^5^, ensuring reasonable quality of fragment spectra. These features are depicted in Figure 2a. Exactly 1805 features were predicted by ML-based classification as prenylated (colored in olive green) across all collision energies (20, 30, 40 eV) used for fragmentation. The overall complexity of the dataset, with a high MS/MS similarity of detected features, is demonstrated in the form of a feature-based molecular network. A high number of isomers was detected for several chemical formulas, including 20 features with the molecular formula C_26_H_32_O_8_ (dark blue crosses in Figure 2a) across a wide retention time range (8.8–19.3 min). All of them were predicted to be prenylated flavonoids, while 14 were manually confirmed as prenylated flavonoids according to the criteria described above, 5 were considered to presumably be prenylated flavonoids, and for 1 of them, evaluation was not possible (due to co-isolation and chimeric spectra). The detailed manual evaluation confirmed the prediction result for the feature with ID 4115 (*m/z* 633.2553, RT 9.8 min). A second, co-eluting feature predicted as a prenylated flavonoid (feature ID 4104, *m/z* 471.2024, chemical formula C_26_H_32_O_8_) was also present. The fragment spectra of ID 4115 indicated a neutral loss of hexose (Appendix A), yielding fragment 471.2024, which confirms the annotation of hexosyl-geranylated flavonoid. Interestingly, CANOPUS did not predict these features to be prenylated. In contrast, feature ID 4115 was classified as a glycosylated flavonoid by CANOPUS, providing complementary insight to ML-based classification results.

In total, 42 features matched reference standards, yielding level 1 confidence annotations serving as key anchors in the molecular networking for confident annotation of many clusters (nodes with red border). For many of those clusters, we observed a high proportion of features predicted as prenylated with a wide variation of retention times, even for ions with the same *m/z*, indicating the presence of numerous isomers. A total of 214 features matched our in-house or public MS/MS libraries with an annotation confidence of level 2, most originating from our in-house library, highlighting its relevance. We therefore assessed the overlap between our compounds and public repositories (without stereochemistry) and compared their spectral similarity (Appendix A). In total, 47 of our compounds were unique, and 17 were not unique in MSnLib, GNPS, and MassSpecGym. The majority of these 47 compounds are geranylated flavonoids. These results illustrate the uniqueness of our compounds and their MS/MS spectra in comparison to public resources.

### 3.3. Isolation and Structure Elucidation of Novel Prenylated Flavonoids

To prioritize isolation efforts, we summarized the number of highly abundant predicted prenylated features based on the ML-based classification described in [48], results of CANOPUS, and manual evaluation as described in Section 2.7.3. To this end, the 200 most abundant features were inspected in detail, and based on the presence of typical adducts (e.g., [M − H]^−^, [2M − H]^−^, [M + HCOONa − H]^−^, [M + HCOO]^−^), occurrence in the same fraction, and feature correlation provided by MZmine the features were grouped into 153 metabolites. The most abundant adduct was selected to represent a particular metabolite, and their number across fractions was summarized along with the number of identified compounds (Figure 3a). The fraction with the highest abundance of the respective prenylated metabolite is reported. The highest number of prenylated metabolites was detected in fractions 6, 7, and 9. However, simultaneously, most of already isolated and identified compounds are also present in these fractions. In general, we observed that fractionation resulted in an increase in the concentration of prenylated compounds compared to the whole chloroform portion of the total extract.

To avoid the re-isolation of already known and described compounds, dereplication with available reference standards was conducted first. Next, a detailed manual evaluation of (partly) coeluting peaks, which could make isolation difficult, was conducted, and these were excluded from further consideration. For the remaining features, representing compounds found to be feasible for isolation, the dereplication results considering HRMS/MS library matching, annotation propagation in molecular networking, and manual evaluation of fragment spectra were checked. This resulted in a selection of potential highly abundant and non-coeluting prenylated targets. To exemplify, several isolation targets are shown in Figure 3b. Based on known fragmentation rules (Figure 3c), the HRMS/MS spectra of the candidate compounds were manually annotated as shown in Figure 3d and Appendix A.

The most abundant metabolites were highlighted in the molecular network, which simplified the view and supported annotation. For instance, feature 8830 at 15.36 min and *m/z* 385.1294 was connected with a node annotated as 6-isopentenyl-3′-O-methyltaxifolin (at RT 15.16 min and *m/z* 385.1294). This revealed an isomeric prenylated dihydroflavonol (compound **1** in Figure 4). Another example was the highlighted feature with ID 15466, which had the first neighbors annotated as tomentodiplacone G and P, which are both flavanones with 4′-hydroxy and 3′-methoxy on the B ring. This shared structural trait explains why the feature ID 15466 clustered with them even though it had a different *m/z* (compound **2** in Figure 4). A similar example consists of the feature 13504, which was clustered with a feature annotated as tomentodiplacone P and also shared the same *m/z* value. Structural similarity of these two ions is evident from the modified cosine score of 0.88 and Figure 4 (compound **4**). In total, five targets were successfully isolated as described in Section 2.5, and their structures were elucidated using UV spectroscopy, HRMS/MS, and 1D and 2D NMR spectroscopy, similar to our previous work [10,49,50,62]. The corresponding data are provided in Appendix A. Briefly, the UV spectra showed maximal values of absorption at ≈295 nm and weak absorption bands at ≈335 nm (Appendix A), which is in line with the absorption spectra of (prenylated) flavonoids [49,50]. Molecular formulas were assigned based on accurate mass within a 5 ppm mass error window (Appendix A). The structures of all compounds were further elucidated by evaluating the NMR data using the ^1^H NMR spectrum together with HSQC, HMBC, COSY, and NOESY experiments.

All of the isolated compounds were prenylated flavonoids identified as a prenylated dihydroflavonol 6-prenyl-4′-*O*-methyltaxifolin (**1**), two flavanones with unmodified geranyl chains, namely 3′-*O*-methyldiplacone (**2**) and 6-geranyl-5,7-dihydroxy-3′,4′-dimethoxyflavanone (syn. 3′,4′-*O*-dimethyldiplacone) (**3**), and two flavanones with modified geranyl chains, namely tomentodiplacone M (**4**) and 3ʹ,4ʹ-*O*-dimethylpaulodiplacone A (**5**) (Figure 4). Two of the compounds (**1** and **5**) were confirmed to be novel flavonoid derivatives described here from a natural source for the first time. These results clearly demonstrate the successful implementation of the proposed workflow for the guided isolation of prenylated flavonoids.

Detailed structure elucidation of the two novel compounds is described in the following paragraphs. Compound **1** was obtained as a yellowish amorphous substance. Its molecular formula was determined by HRMS to be C_21_H_22_O_7_ based on the presence of the molecular ions [M − H]^−^ and [M + H]^+^ at *m/z* 385.1291 and 387.1441, respectively (calculated for C_21_H_21_O_7_^−^, 385.1293 and for C_21_H_23_O_7_^+^, 387.1438). The ^1^H NMR spectrum displayed two doublets at *δ*_H-2,3_ 4.95 and 4.51 ppm, typical for a dihydroflavonol skeleton with coupling constant *J* = 11.9 Hz, indicating a *trans* arrangement of H-2 and H-3 [63]. The singlet at *δ*_H_ 5.98 ppm was assigned to H-8 because of its HMBC correlations to C-6, C-7, C-9, and C-10, and a weaker correlation to C-4, as described previously [10,49,50]. Carbon C-6 was assigned as the connection of the prenyl chain to the flavonoid skeleton based on the HMBC correlations of H-1′′ with C-5, C-6, C-7, and a NOESY correlation of H-1′′ with OH-5 [49,50]. The phenolic ring B of the flavonoid skeleton was substituted with methoxy and hydroxy groups, but chemical shifts were different compared to the 3′-methoxy-4′-hydroxy arrangement of compounds **2** and **4** and also previously isolated compounds [49,50]. Using DMSO-*d*_6_, it was not clear whether the methoxy group showed an NOE correlation with H-2′ or H-5′, as these signals overlapped. Therefore, this compound was also measured in CDCl_3,_ and there were three separate signals at *δ*_H_ 6.90, 7.01, and 7.13 ppm in the ^1^H NMR spectrum (Appendix A). The methoxy signal at *δ*_H_ 3.91 ppm showed the NOE correlation with the *ortho*-coupled aromatic proton at *δ*_H_ 6.90 ppm (H-5′), indicating a 3′-hydroxy-4′-methoxy arrangement of ring B for compound **1**, which was also described previously in *P. tomentosa* [64]. NMR data further revealed the signals for an unmodified prenyl chain consisting of two methyl singlets (*δ*_H-4′′,5′′_ 1.75 and 1.81 ppm), one methylene (*δ*_H-1′′_ 3.35 ppm), and one methine group (*δ*_H-2′′_ 5.24 ppm) with coupling constant values *J* = 7.3 Hz, together with one quaternary carbon of the double bond (*δ*_C_ 135.7 ppm). This compound was described for the first time from plant material, and because of its similarity with previously isolated 6-isopentenyl-3′-*O*-methyltaxifolin [63], it was named 6-prenyl-4′-*O*-methyltaxifolin.

Compound **5** was obtained as a yellowish amorphous substance. The molecular formula was determined by HRMS to be C_27_H_32_O_7_ based on the presence of the molecular ions [M − H]^−^ and [M + H]^+^ at *m/z* 467.2068 and 469.2226, respectively (calculated for C_27_H_31_O_7_^−^, 467.2075 and for C_27_H_33_O_7_^+^, 469.2221). A very prominent [M − H_2_O + H]^+^ adduct in full scan spectra was found to be diagnostic for 2′′-hydroxygeranylated flavonoids, which can be used for the indication of hydroxy position on the geranyl side chain. The UV spectrum and NMR data (Table 1) revealed a flavanone skeleton [10,49,50]. Signals for three aromatic protons at *δ*_H_ 6.92 (d, *J* = 8.2 Hz), 6.97 (dd, *J* = 1.8, 8.2 Hz), and 7.07 (d, *J* = 1.8 Hz) were assigned to the 3′,4′-disubstituted ring B together with two methoxy signals at *δ*_H_ 3.72 and 3.73 ppm. NMR analysis further revealed a geranyl chain modified by oxidation with the formation of a terminal double bond. This modification was reported previously [10,49,50]. HMBC and COSY correlations confirmed this assignment, but unfortunately, a too low amount was obtained (0.8 mg) to clearly see all correlations, as in cases of other isolated compounds. This compound was isolated from a natural source for the first time and structurally assigned as shown. Because its structure is similar to that of paulodiplacone A [49], and considering its 3′,4′-dimethoxy ring B, it was named as 3′,4′-*O*-dimethylpaulodiplacone A (**5**).

## 4. Discussion

This study demonstrates that integrating extraction and fractionation protocols with LC-UV-HRMS/MS analysis and customized machine-learning models to classify the LC-HRMS/MS spectra can be used to significantly enhance the targeted isolation of prenylated flavonoids. The discovery of previously unreported compounds is exemplified by samples from the princess tree *P. tomentosa*. Paulownia trees are fast-growing and moisture-resistant, and they have been recognized as a potential bioenergy crop, construction material, and medicinal plant in traditional Chinese medicine. Particularly, fruits are used for wound healing, tonsillitis, bronchitis, and bacterial infections [65]. The immature fruits contain viscous and sticky material on the surface, for which numerous geranylated flavonoids have been reported [49,64]. Several *in vitro* studies have also demonstrated potent anti-inflammatory properties of the isolated geranylated flavonoids from *P. tomentosa* fruit extracts [10,50].

In the pursuit of novel natural products, various dereplication strategies have been employed, as outlined by Hubert et al. [66]. Our approach builds on the established bioactivity of prenylated flavonoids, allowing us to bypass traditional bioactivity-guided fractionation and instead directly target prenylated flavonoids through LC-HRMS/MS combined with precise manual dereplication. Although presented here as a targeted strategy focusing on a specific compound class, this workflow is not restricted to downstream data analysis, and the generated dataset may also be valuable for researchers not primarily interested in prenylated compounds. However, the extraction conditions should be carefully considered.

The proposed LLE extraction steps, starting with hexane, remove non-polar hydrophobic compounds, including lipids and waxes. The subsequent extraction with chloroform is well suited to capture the whole range from rather hydrophobic to more polar prenylated flavonoids, while at the same time reducing sample complexity. The extraction step critically influences acquired metabolite profiles, which in turn can affect the performance of ML-based classification models. A similar situation can arise when extending the workflow to other plant species or compound classes. To mitigate this, we included a wheat methanolic extract in the training phase to provide a broad set of non-prenylated metabolites, thereby increasing the diversity of metabolites to be recognized by the models [48]. Overall, the quality of predictions can be influenced by extraction protocols and analyzed samples (tissues and plant species) and needs to be considered critically, even though the models were carefully trained and validated on reference standards [48]. True validation within the biological system under investigation is often impossible due to the lack of ground truth (i.e., lack of confirmation with reference standards). Instead, results can be compared against alternative reference points, such as manual annotation or other classification/annotation tools (e.g., CANNOPUS or library matching). While this represents a proxy rather than genuine validation, it still provides important insights into classification performance.

Fractionation results in fewer co-eluting compounds and improves chromatogram peak resolution and sensitivity, which affects LC-HRMS analysis and isolations with semi-preparative HPLC. Only a few features were found to be more abundant in the whole chloroform extract. This can be reasoned by splitting the amount of a single compound into two fractions, or degradation and loss of some constituents during the fractionation process. Although fractionation simplifies the composition of individual fractions compared to the raw extract for isolation, it also leads to an increase in the abundance of minor constituents due to the associated sample enrichment. This can lead to higher HRMS/MS dataset complexity, which is also reflected in high numbers of interconnected nodes in the molecular network derived from chloroform fractions. The high number of detected isomers can partially be explained by different chemical modifications (e.g., hydroxy and methoxy groups) and their position at both the flavonoid core structure and geranyl side chain. Moreover, the presence/absence of a double bond in flavone/flavanone pairs can be matched with the cyclization of the geranyl side chain, accompanied by hydrogen loss. The early eluting prenylated flavonoids were likely formed in the electrospray ion source of the mass spectrometer, as evidenced by an example of a simultaneously glycosylated and geranylated flavonoid. The attachment of highly polar moieties, such as sugar, sulfate, or phosphate groups, results in earlier elution of prenylated flavonoids. However, due to in-source fragmentation, the intact molecular ion is not observed, and only the remaining geranylated flavonoid fragment is detected. The simultaneously glycosylated prenyl-flavonoids are of particular interest for drug development, as they can combine high biological activity with enhanced solubility and potential for improved bioavailability [67,68]. The targeted isolation of glycosylated prenyl-flavonoids from plant material may require modifications to our current separation methods to achieve higher extraction and separation efficiencies. The custom-trained machine learning models efficiently predicted prenylated flavonoids even from a small set of reference spectra for that particular substance class. This helped reduce the vast chemical space of the LC-HRMS dataset and facilitated targeting particularly interesting early-eluting prenylated constituents, while taking advantage of the complementarity with CANOPUS. Due to its adaptability across collision energies and ionization modes and its user-friendly implementation in the custom-built AnnoMe Python package [48], this approach promises widespread use in untargeted metabolomics and natural products discovery. Compared to other approaches utilizing only full scan MS1 data, where different filters based on, e.g., mass defect, Van Krevelen diagrams, or retention times are applied, our strategy uses MS/MS fragment spectra, which contain structural information and also enable better selectivity. However, since our approach relies on the informative MS/MS data, it can be problematic to generate enough fragment-rich MS/MS spectra across a chromatographic peak, especially for low-abundant constituents. Moreover, while ML-based models can evaluate MS/MS data in a high-throughput way, they rely on the availability of MS/MS reference data for their training and are focused on pre-defined specific structural compound class(es) (e.g., prenylated flavonoids). Broader applicability to further metabolite classes will require further library expansion, model re-training, and validation. The choice of adequate fragmentation conditions and ionization polarity is crucial, as suited conditions may vary across compound classes and will affect the classification outcome. These points should therefore be addressed on an individual basis to generate information-rich data for the classification models, as demonstrated [48]. Another limitation of our workflow is the variation in MS/MS spectra across different instrument types [69], which may limit the broader applicability of the optimized models. Therefore, it should be noted that widespread application of the used models requires additional testing and, if necessary, further optimization beyond the applied Orbitrap platforms in combination with HCD collision energies.

The in-house MS/MS library provided high-confidence annotations and, together with manual spectra inspection and evaluation, facilitated reliable dereplication. The ML-classification models can supplement and enhance the efficiency of manual annotation and screening; however, an ML-based procedure alone cannot establish such confidence. While SIRIUS annotations were often correct, they were also sometimes incorrect at the same time, e.g., suggesting structures with two prenyl (C5) units instead of a single geranyl (C10) moiety, or for compound **5**, the wrong number of substituents (dimethoxy and hydroxy) were reported on ring B. All of this can be resolved from MS/MS spectra based on fragmentation mechanisms, providing a detailed structure description. Our results also emphasize the added value and uniqueness of the spectral library with a significant number of prenylated structures compared to existing public resources. Lack of fragmentation spectra for geranylated flavonoids during SIRIUS development can explain its annotation inaccuracy, or the molecular fingerprints cannot cover structural traits in such detail. Problematic could also be the assigned ClassyFire class, which does not contain the “prenyl” keyword. Therefore, the custom-built ML-based binary classifier offers a more user-friendly and tunable approach. Moreover, our MS/MS data will be made publicly available as a valuable resource to support annotation efforts and advance machine learning applications [39]. While three prenylated targets, compounds **2**–**4**, were isolated previously [10,64,70], they were absent from the in-house library of geranylated flavonoids, highlighting the challenge of resolving positional isomers using current fragmentation techniques (e.g., distinguishing OH substitution at positions C-2″, C-7″, or C-8″ on the geranyl moiety), as well as the limitations of manual annotations. The manual spectra evaluation is affected mainly by the operator’s expertise and can vary over time, but in principle can be reliable, especially in combination with software applying fragmentation rules from literature [40]. Therefore, implementation of such a tool could be the next step in improving the presented workflow.

Prenylated flavonoids are increasingly recognized as bioactive natural products with diverse physiological roles and biological activities [65,71]. These compounds were reported to have antioxidant [6], anti-inflammatory [50], antimicrobial [62], and metabolic regulatory effects [7,8], which may contribute to plant defense (against UV radiation, insects and herbivores, or microbes) as well as exhibit potential health benefits in humans. The prenyl/geranyl group is thought to enhance lipophilicity and membrane affinity, thereby increasing bioavailability and biological potency [11]. While our study focused on the annotation and isolation of the detected prenylated flavonoids, future directions should include bioactivity-guided validation of the identified and isolated compounds to link chemical features with biological effects. Particularly intriguing are the anti-inflammatory properties of polyphenols, which can be helpful to manage inflammatory diseases, including neurodegenerative disorders, arthritis, and metabolic conditions [72]. The bioconversion of prenylated flavonoids towards their glycosides results in improved solubility, an interesting property that could support novel drug development strategies in the future [68].

Future efforts could integrate a rule-based annotation tool or retention time prediction models to provide reliable dereplication and to improve isomer differentiation. The number of reference spectra used to train the classification models can be expanded to provide more robust results, and additional strategies involving full scan MS1 data, for example, Van Krevelen diagrams, can be implemented as extra validation criteria for the classification of prenylated flavonoids. Expanding the scope to other prenylated compounds could further enhance natural product discovery workflows but would require further retraining and validation of the models. Potential users are advised to ensure sufficient quality and information content of HRMS/MS fragment spectra and proper validation of the classifier.

## 5. Conclusions

This study successfully demonstrates the integration of sample extraction and fractionation protocols with advanced analytical techniques, with an AI-enhanced workflow for the efficient isolation and characterization of prenylated flavonoids from *P. tomentosa* fruit extracts. By combining LC-UV-HRMS/MS analysis with custom-trained machine-learning models, the workflow effectively annotated the constituents of complex plant extracts, enabling the prioritization of fractions and compounds for targeted isolation. This approach led to the structural elucidation and discovery of five prenylated flavonoids, including two novel compounds, 6-prenyl-4′-*O*-methyltaxifolin (**1**) and 3′,4′-*O*-dimethylpaulodiplacone A (**5**), isolated from a natural source for the first time.

A key strength of this workflow lies in the in-house developed high-quality MS/MS library, which includes spectra of unique geranylated flavonoids. This library can also serve as a valuable resource for compound annotation and dereplication, and is also available for the future development of AI tools.

The workflow is versatile and can be adopted for screening prenylated flavonoids not only in natural product research but also in metabolomics, given the bioactive potential of these compounds and their roles in plant-environment interactions. Furthermore, the approach can be extended to different collision energies, polarities, and potentially other classes of prenylated constituents upon further training and validations, paving the way for broader applications in natural product discovery and metabolomics.

## Figures and Tables

**Figure 1 metabolites-15-00616-f001:**
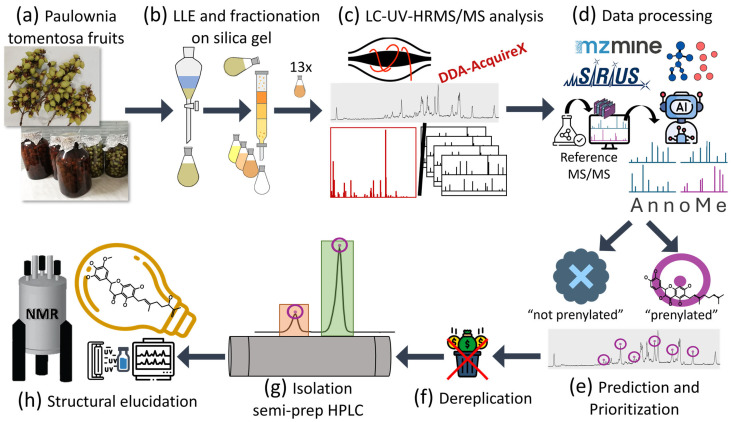
Workflow scheme utilizing LC-UV-HRMS/MS and ML-based classification for high-throughput prediction of prenylated compounds to characterize plant extracts and streamline isolation of most likely bioactive prenylated flavonoids. (**a**) Maceration in ethanol to yield crude extract; (**b**) Liquid-liquid extraction and fractionation of chloroform portion; (**c**) LC-UV-HRMS/MS analysis of chloroform fractions; (**d**) Data processing using MZmine, SIRIUS, molecular networking, and high-throughput prediction of prenylated flavonoids by ML-based classification; (**e**) Prediction and prioritization of fractions and prenylated constituents for isolation; (**f**) Dereplication; (**g**) Guided isolation with semi-preparative HPLC system; (**h**) Structure elucidation using UV spectroscopy, LC-HRMS, and 1D and 2D NMR.

**Figure 2 metabolites-15-00616-f002:**
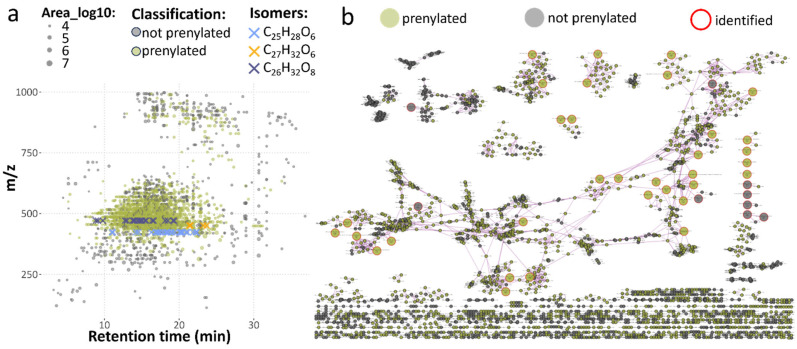
Overview of all detected constituents across 13 chloroform fractions. (**a**) Feature map with ions characterized by retention time and *m/z*; those predicted as prenylated flavonoids have an olive-green color, while the rest are gray. (**b**) Feature-based molecular network based on negative polarity, HCD 40 eV, cosine > 0.7, and a minimum of 4 matching fragments, showing high connectivity within the network associated with a high MS/MS spectra similarity of detected features with identified compounds displayed as bigger nodes with red borders.

**Figure 3 metabolites-15-00616-f003:**
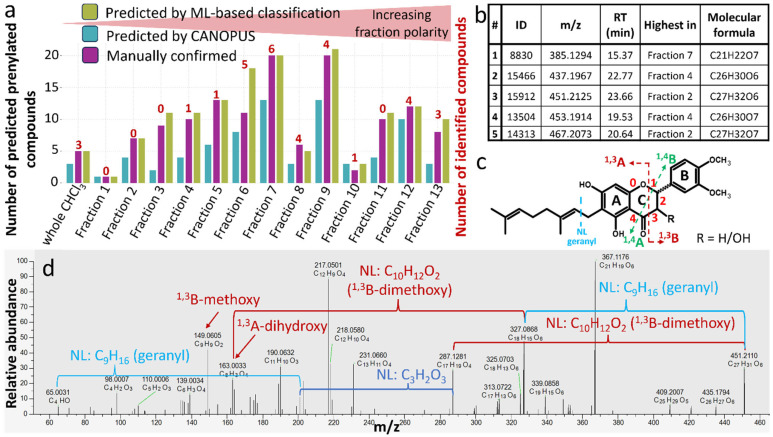
Prioritization of fractions for isolation and detailed characterization of suitable isolation targets. (**a**) Number of prenylated features within the top 200 most abundant features across fractions, predicted by ML-based classification, CANOPUS, or manually confirmed, together with the number of already isolated and identified compounds from previous work included in our in-house MS/MS library. (**b**) Selection of suitable isolation targets. (**c**) Fragmentation scheme of geranylated flavonoids. (**d**) Manually annotated MS/MS spectra of feature ID 15912.

**Figure 4 metabolites-15-00616-f004:**
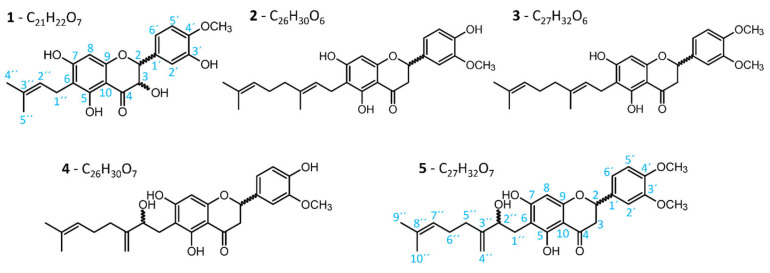
Structures of compounds **1**–**5** isolated from the fruits of *P. tomentosa* utilizing the presented workflow.

**Table 1 metabolites-15-00616-t001:** NMR spectroscopic data for 6-prenyl-4′-*O*-methyltaxifolin (**1**) (400 MHz, CDCl_3_) and 3′,4′-*O*-dimethylpaulodiplacone A (**5**) (400 MHz, DMSO-*d*_6_).

	6-Prenyl-4′-*O*-methyltaxifolin (1)			3′,4′-*O*-Dimethylpaulodiplacone A (5)		
Position	*δ*_C_, Type	*δ*_H_ (*J* in Hz)	HMBC	*δ*_C_, Type	*δ*_H_ (*J* in Hz)	HMBC
2	83.3, CH	4.95, d (11.9)	3, 4, 1′, 2′, 6′	78.5, CH	5.30, m	n. o.
3	72.4, CH	4.51, d (11.9)	2, 4, 1′	42.8, CH_2_	2.57, m	4
					3.12, m (ov *)	2, 4
4	196.0, C			194.3, C		
5	160.6, C			161.6, C		
6	107.6, C			107.1, C		
7	164.8, C			161.6, C		
8	96.0, CH	5.98, s	4, 6, 7, 9, 10	96.4, CH	5.65, s	6, 7, 9, 10
					5.66, s	
9	161.1, C			161.1, C		
10	100.6, C			100.2, C		
1′	129.3, C			132.3, C		
2′	113.6, CH	7.13, d (2.3)	2, 4′, 6′	111.1, CH	7.07, d (1.8)	2, 4′, 6′
3′	145.9, C			149.3, C		
4′	147.4, C			149.4, C		
5′	110.6, CH	6.90, d (8.2)	1′, 3′	112.1, CH	6.92, d (8.2)	1′, 3′
6′	119.9, CH	7.01, dd (2.3, 8.2)	2, 2′, 4′	119.5, CH	6.97, dd (1.8, 8.2)	2, 2′, 4′
1′′	21.2, CH_2_	3.35, d (7.3)	5, 6, 7, 2′′, 3′′	29.9, CH_2_	2.50, m	5, 6, 7, 2′′, 3′′
					2.66, m	5, 6, 7, 2′′, 3′′
2′′	121.2, CH	5.24, t (7.3)	6, 1′′, 4′′, 5′′	74.7, CH	4.06, m	n. o.
3′′	135.7, C			152.7, C		
4′′	25.9, CH_3_	1.75, s	2′′, 3′′, 5′′	108.3, CH_2_	4.60, br s	2′′, 3′′, 5′′
					4.81, br s	2′′, 3′′, 5′′
5′′	17.9, CH_3_	1.81, s	2′′, 3′′, 4′′	31.5, CH_2_	2.05, m (ov)	3′′, 6′′
6′′				26.5, CH_2_	2.05, m (ov)	5′′
7′′				125.2, CH	5.09, m	n. o.
8′′				131.2, C		
9′′				26.0, CH_3_	1.61, s	7′′, 8′′, 10′′
10′′				18.2, CH_3_	1.54, s	7′′, 8′′, 9′′
OH-5		11.53, s	5, 6, 10		12.58, s	5, 6, 10
					12.59, s	
MeO-3′				56.1, CH_3_	3.73, CH_3_	3′
MeO-4′	56.1, CH_3_	3.91, s	4′	56.1, CH_3_	3.72, CH_3_	4′

n. o.—not observed; ov—signals overlapped; ov *—signal overlapped with residual H_2_O signal; carbons were assigned based on the Heteronuclear Single Quantum Correlation (HSQC) and Heteronuclear Multiple Bond Correlation (HMBC) data.

## Data Availability

The raw LC-UV-HRMS/MS data generated and analysed during the current study are available in the MASSIVE repository, https://massive.ucsd.edu/ProteoSAFe/dataset.jsp?task=f4420d487b344b9eabe6fa3f65695fee, accessed on 12 September 2025. The in-house MS/MS library is accessible at ZENODO: https://doi.org/10.5281/zenodo.16762591.

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
