# Peer review of "Targeted Isolation of Prenylated Flavonoids from *Paulownia tomentosa* Fruit Extracts via AI-Guided Workflow Integrating LC-UV-HRMS/MS"

_metabolites, 2025, doi:10.3390/metabo15090616_

Round 1
Reviewer 1 Report
Comments and Suggestions for Authors
The manuscript presents a novel approach and addresses the key challenge in manual characterisation analysis in metabolomics. The results are quite interesting. I have a few suggestions/comments (given below and attached in the manuscript)
Introduction: Describe the research gap
The author must include the limitations of the study/AI-based proposed model in the conclusion/discussion. AI can supplement and enhance the efficiency of manual screening; however, an entirely AI-based procedure could not establish confidence.
The model must be trained for different biological species since the nature has tremendous diversity in metabolites and a lot is still undiscovered. The matrix also contributes significantly to extracting the specific compound and the extraction system too. Though the author have carefully included in the study, they must have discussed their relevance with different solvent/extraction systems or the effect of the matrix.

Author Response
We thank the reviewer for the thoughtful and constructive feedback, which helped us improve the manuscript. Please find a detailed response in the attached file.

Reviewer 2 Report
Comments and Suggestions for Authors
- Please elaborate on the design and training of the AI model, including the dataset, validation metrics (e.g., accuracy, precision), and how it improves upon traditional selection methods. A workflow diagram would be highly beneficial.
- Include information on the origin, collection conditions, and any efforts to minimize seasonal or environmental variability in the chemical profiles of P. tomentosa fruits.
- The use of PCA and molecular networking is appreciated, but further discussion is encouraged on how these analyses led to new findings rather than reconfirming known metabolite clusters.
- Consider elaborating on the potential biological roles of identified prenylated flavonoids, and suggest future directions such as bioactivity-guided validation, in vitro/in vivo studies, or applications in functional food or drug development.
- A comparative discussion of your AI-guided approach versus traditional extraction or dereplication strategies would help underscore the practical benefits of your method.
Author Response

(The authors gave the same response as above.)

Reviewer 3 Report
Comments and Suggestions for Authors
This manuscript describes a new workflow combining machine learning and LC-HRMS/MS for natural product annotation and isolation, which is very interesting. The expirements are well- designed and results are analysed properly. There is a substantial amount of work involved in isolating the target compounds and performing NMR analysis.I found this paper very enjoyable to read. Here are my comments and suggestions: In introduction or discussion section, the authors should explain why they choose Paulownia tomentosa fruit to extract prenylated flavonoids. In line 50, they said prenylation can enhance bioactivies. But what kinds of bioactivities do they talk about? In the discussion, it would also be worthwhile to mention any limitations of this workflow.
Author Response

(The authors gave the same response as above.)
